# Single-Cell Dissection of the Serrated Pathway: Cellular Heterogeneity and Genetic Causality in Colorectal Cancer

**DOI:** 10.3390/ijms26157187

**Published:** 2025-07-25

**Authors:** Ming Cen, Yunhan Wen, Zhijun Feng, Yahai Shu, Chuanxia Hu

**Affiliations:** 1Jiangsu Province High-Tech Key Laboratory for Bio-Medical Research, School of Life Sciences and Technology, Advanced Institute for Life and Health, Southeast University, Nanjing 210096, China; 220223590@seu.edu.cn; 2Guangzhou National Laboratory, Guangzhou 510005, China; yunhanwen2014@163.com (Y.W.); fengzhj18@sina.com (Z.F.); 3Guangzhou Institutes of Biomedicine and Health, Chinese Academy of Sciences, Guangzhou 510005, China

**Keywords:** serrated epithelial cell, colorectal cancer, tumor progression, single-cell transcriptomics, Mendelian randomization

## Abstract

The serrated pathway represents a significant route to colorectal cancer (CRC), accounting for approximately 15–30% of cases, yet the specific epithelial cell subpopulations driving this pathway remain poorly understood. This study explores the causal relationship between serrated epithelial cells and CRC risk using single-cell transcriptomics and Mendelian randomization (MR). Publicly available single-cell RNA sequencing data were utilized to analyze epithelial cell subpopulations in CRC, focusing on specific serrated cells (SSCs). By integrating genome-wide association study data, MR was employed to assess the causal relationship between gene expression patterns and CRC risk. The study found that an increase in SSCs is closely associated with CRC progression. MR analysis revealed a significant correlation between expression changes in specific genes, such as *IER3* in SSCs, and CRC risk (*p* < 0.05). Functional analyses indicated that *IER3* may promote malignancy by regulating cell proliferation, adhesion, and immune evasion. Several genetic loci related to SSC gene expression were identified and validated for CRC risk association. This study demonstrates the significant role of serrated epithelial cell subpopulations in CRC development, particularly through key genes such as *IER3*, providing new perspectives for understanding CRC pathogenesis and future therapeutic strategies.

## 1. Introduction

Colorectal cancer (CRC) remains one of the most common malignancies worldwide, with persistently high incidence and mortality rates. According to Global Cancer Statistics 2020, CRC ranks as the third most prevalent cancer and the second leading cause of cancer-related deaths [1]. The situation is particularly concerning in China, where CRC has become the second most common cancer, with approximately 510,000 new cases annually and 240,000 deaths, representing a significant public health challenge [2]. Despite advancements in diagnostic and therapeutic techniques, the overall prognosis for CRC remains suboptimal, particularly for patients diagnosed at advanced stages, who have significantly lower five-year survival rates. Therefore, investigating the pathogenesis of CRC, identifying novel biomarkers, and discovering new therapeutic targets have become critical endeavors in cancer research.

In recent years, serrated lesions have gained significant attention as precursors to CRC [3,4]. These lesions, characterized by their distinctive saw-tooth epithelial pattern, include hyperplastic polyps (HP), sessile serrated lesions (SSL, also known as sessile serrated adenomas/polyps or SSA/P), and traditional serrated adenomas (TSA) [5,6]. Among these, SSL and TSA have been recognized for their malignant potential, with approximately 15–30% of sporadic CRCs developing through the serrated pathway [7,8,9]. This pathway represents a distinct route of carcinogenesis that differs from the conventional adenoma-carcinoma sequence and Lynch syndrome pathway [10,11]. Molecularly, the serrated pathway is characterized by several key alterations, including MAPK pathway activation and varying degrees of CpG island methylator phenotype (CIMP) [12,13,14]. Two primary serrated carcinogenesis models have emerged: one involving BRAF mutation with high-level CIMP leading to either high microsatellite instability (MSI-H) or microsatellite stable (MSS) CRC, and another featuring KRAS mutation with low-level CIMP progressing to MSS-type CRC [15,16]. The former is strongly associated with the sessile serrated pathway, while the latter relates more to the traditional serrated pathway [17].

Despite growing recognition of serrated lesions’ importance in CRC development, these lesions remain challenging to detect clinically. Studies have reported misdiagnosis rates as high as 59% for proximal colon serrated lesions [18]. This high rate of missed diagnoses may contribute significantly to interval cancers—those occurring after a negative colonoscopy—which are more frequently observed in the proximal colon and associated with the serrated pathway. The difficulty in detecting these lesions, coupled with their significant contribution to CRC burden, underscores the urgent need for improved diagnostic approaches. While substantial progress has been made in understanding the clinical features and molecular characteristics of serrated lesions [19,20,21], a critical knowledge gap persists regarding their specific cellular components and heterogeneity. This limited understanding of the cellular landscape within serrated lesions represents a significant obstacle to developing more effective detection strategies and targeted interventions.

With the advancement of single-cell RNA sequencing (scRNA-seq) technology, an increasing number of studies are exploring cellular heterogeneity within the tumor microenvironment (TME) [22,23], examining the roles of different cell subpopulations in tumor development at the single-cell level [24,25]. This technology offers unprecedented opportunities to identify and characterize specific serrated cells (SSCs) in the colorectal context, potentially revealing new insights into the pathogenesis of serrated pathway CRCs. Mendelian randomization (MR) analysis, a genetics-based research method, effectively assesses causal relationships between genes and diseases. By using genetic variants as instrumental variables, MR analysis simulates the grouping process of randomized controlled trials, significantly reducing confounding factors common in traditional epidemiological studies and enhancing the credibility of causal inferences [26,27]. In cancer research, MR analysis has been widely applied to explore the causal relationships between gene expression and cancer risk, providing new theoretical bases for disease prevention and treatment [28].

In this study, we utilized scRNA-seq to analyze CRC samples, identify specific cell subpopulations related to serrated lesions, and characterize their defining genes. Subsequently, by combining this information with genome-wide association study (GWAS) data, we will employ MR analysis to verify the causal relationships between these characteristic genes and CRC. This integrative approach will not only provide new insights into the heterogeneity of CRC but also potentially offer novel targets and biomarkers for precision medicine practices, ultimately contributing to improved diagnosis, prevention, and treatment strategies for serrated pathway CRC.

## 2. Results

### 2.1. Single-Cell Atlas and Cellular Functional Exploration in CRC

Clustering analysis of human CRC scRNA-seq at a resolution of 0.8 identified 20 distinct cell clusters (Appendix A). Using representative marker genes for cell types, we identified prevalent cell populations, including B cells, endothelial cells, epithelial cells, monocytes, NK cells, smooth muscle cells, T cells, and tissue stem cells (Figure 1A and Appendix A). Notably, epithelial cells were significantly increased in CRC tissues, whereas immune cells, including T cells, were markedly reduced (Figure 1B). Distinct cell groups exhibited unique gene expression profiles (Figure 1C). Functional enrichment analysis across these cell types demonstrated their inherent characteristic functions, such as cell adhesion among epithelial cells, activation functions in immune cells, and contributions by stromal cells (smooth muscle and endothelial cells) to the extracellular matrix and angiogenesis (Figure 1C). These findings underscore the critical role of the tumor microenvironment in CRC. Specifically, the increase in epithelial cells correlates closely with tumor growth and invasiveness, while the reduction in immune cells may reflect tumor immune evasion, whereby tumors reduce immune cell infiltration to evade host immune surveillance. By integrating single-cell transcriptomic data and functional enrichment analysis, we have not only mapped the cellular heterogeneity of CRC but also revealed the complexities of intercellular interactions and signaling networks. These insights deepen our understanding of the tumor microenvironment’s role in CRC progression and highlight potential targets for future therapeutic strategies.

### 2.2. Characteristics of CRC Epithelial Cell Subgroups

The analysis revealed 12 initial epithelial cell clusters within CRC (Figure 2A), with significant variability in the proportion of each cluster across different samples (Figure 2B). Functional enrichment analysis of these clusters showed that, in addition to possessing typical epithelial cell genes (Appendix A) and functional characteristics such as cytokine interactions, they also displayed features associated with immune cell functions, such as granulocyte migration and lymphocyte-mediated immunity (Figure 2C). Based on shared molecular signatures and functional profiles, these 12 clusters were consolidated into four distinct epithelial cell subgroups: specific serrated cells (SSC), absorptive cells (ABS), transit amplifying cells (TAC), and goblet cells (GOB) (Figure 3A). Compared with normal epithelial cells (CT group), there was a noticeable increase in SSCs and a decrease in ABS cells in tumor patients (Figure 3A). These cells also exhibited distinctive gene characteristics from each other (Figure 3B). This evidence indicates that the features of epithelial cell subgroups in CRC are closely linked with tumor progression and immune modulation. The increase in SSCs (stem cell-like cells) may relate to tumor self-renewal and enhanced invasiveness, while the decrease in ABS could impact intestinal function and the stability of the tumor microenvironment. These findings also highlight the complexity involved in the development and progression of CRC.

### 2.3. Epithelial Cell Trajectory Evolution and Interactions with Other Cell Types

Trajectory analysis of epithelial cell subgroups revealed three distinct developmental nodes and two divergent developmental trajectories among all epithelial cells (Figure 4A). Based on these developmental nodes, all epithelial cells were classified into seven states (Figure 4B). The distribution characteristics of cell populations within each state showed that SSC and TAC cells were positioned at the beginning of the differentiation trajectory, with developmental endpoints predominantly consisting of ABS cells on one end and GOB cells on the other (Figure 4C). The density of cells within different developmental stages confirmed the dominant cell types at each trajectory position (Figure 4D). Notably, a significant distribution of GOB cells was observed along the developmental trajectory toward ABS cells. Subsequently, interaction analyses using CellChat assessed the interactions between epithelial cell subgroups and various cell types, revealing enhanced signaling interactions, particularly between SSC and endothelial cells and TAC and endothelial cells, both in terms of quantity (Figure 4E) and intensity (Figure 4F). Given the significant reduction in ABS cells in tumor tissues, we focused on evaluating the interaction signals between ABS and other cells, identifying the MDK–NCL signal as a primary communication pathway among these cells, playing a significant role in regulating interactions across different cell types (Figure 4G). These findings indicate that different developmental states of epithelial cells significantly impact their functionality and communication patterns. Especially within the tumor microenvironment, SSCs, positioned at the origin of the developmental trajectory, possess a high capacity for self-renewal and multilineage differentiation, crucial for maintaining epithelial tissue stability and repair. They likely influence tumor development and treatment response by facilitating transitions between different cell states. In this study, interaction analyses highlighted strengthened signaling interactions between SSCs and endothelial cells, which may involve the regulation of tumor angiogenesis and the shaping of the tumor growth environment. This suggests that SSCs not only play a pivotal role in maintaining normal epithelial tissue functions but also impact pathological processes through dynamic interactions with the surrounding microenvironment. The reduction in ABS cells might lead to a reorganization of the intercellular communication network, thereby affecting tumor growth and development. Additionally, the distribution of GOB cells across both developmental trajectories suggests they may play key roles in cell differentiation and pathological state transitions. These insights provide potential molecular targets for developing treatment strategies targeted at specific cell subgroups, which could significantly enhance treatment efficacy and prognosis.

### 2.4. Gene Characterization in SSCs

Differential analysis between SSCs and non-SSCs across all cell types resulted in a list of differential genes (Appendix A). Further differential analysis between SSC and non-SSC epithelial cells yielded a list of SSC-specific differential genes (Appendix A). We subsequently extracted common differential genes from these datasets, totaling 246 genes (Figure 5A, Appendix A). The PPI network confirmed the existence of varying degrees of interaction between these genes (Figure 5B). Molecular component analysis (Figure 5C) indicated that these genes are primarily involved in focal adhesion, extracellular matrix, and melanosomes. From a molecular function perspective (Figure 5D), these genes are predominantly involved in cadherin binding, cytokine activity, and cell–cell adhesion. In terms of biological processes (Figure 5E), these common differential genes are primarily involved in wound healing, leukocyte migration, and cell–cell adhesion. Detailed results of the GO analysis for these genes are shown in Appendix A. Pathway enrichment analysis (Figure 5F) demonstrated that these genes are mainly involved in pathways such as the PI3K–Akt signaling pathway, focal adhesion, and HIF-1 signaling pathway, which are related to metabolism (Appendix A). These findings suggest that these common differential genes may play critical roles in specific biological processes and signaling pathways, particularly in cell adhesion, tissue repair, and immune response. Moreover, the significant enrichment of these genes indicates their involvement in SSCs’ adaptation and response mechanisms within specific pathological environments, providing new perspectives and potential therapeutic targets for further understanding the role of SSCs in disease onset and progression.

### 2.5. Shared Differential Gene eQTLs and Colorectal Cancer Risk

Using summarized eQTLGen 2019 data from the OpenGWAS database, we extracted genetic locus information related to the 246 shared differential genes, identifying 668 significant genetic loci corresponding to 180 genes (Appendix A). Subsequent MR analysis using the IVW method indicated positive correlations between *RNASET2*, *ANXA4*, and *TMBIM1* and CRC risk, while *RAB11A*, *IER3*, *SCARB2*, *ATF3*, and *CXCL8* were negatively correlated with CRC risk (Figure 6A). Detailed results of the MR analysis are presented in Appendix A. Further validation using CRC single-cell data confirmed varied expression levels of these eight genes across different cell types, particularly within immune cells; *ANXA4* was primarily expressed in epithelial cells, while *RNASET2* was mainly found in monocytes (Figure 6B). Functional enrichment analysis of these genes revealed their involvement in key biological processes such as the regulation of the extrinsic apoptotic signaling pathway, cytokine receptor binding, and *CXCR* chemokine receptor binding (Figure 6C). Additionally, these genes were implicated in biological pathways related to organ development, such as endosome organization, vacuolar lumen, and late endosome. These findings suggest that these genes may play significant roles in the onset and progression of CRC, particularly through the regulation of extrinsic apoptotic signaling pathways and cytokine receptor interactions. The differential expression of these genes in specific cell types further indicates their potential roles in modulating the CRC microenvironment, immune evasion, or cancer cell growth, offering new insights into CRC’s molecular mechanisms and potential targets for precision therapy.

### 2.6. Switch Genes and Their Impact on Colorectal Cancer Epithelial Cells

Using GeneSwitches analysis, we qualitatively studied the “switch” roles of eight genes prominently featured in the MR analysis within epithelial cells. Results (Table 1) indicated that *IER3*, *CXCL8*, *ANXA4*, *ATF3*, and *RAB11A* genes were activated during the early stages of cell development, while *SCARB2*, *RNASET2*, and *TMBIM1* were activated in later stages, with IER3 having the highest explanatory power (Table 1, estimated value = 0.32). Additionally, other types of genes also demonstrated significant “switch” effects during CRC epithelial cell development (Figure 7A), such as surface protein molecules *SNRPD2*, *SLC12A2*, *XRCC6*, and *CEACAM1*, as well as potential transcription factors *ETS1*, *PRDM1*, *EGR1*, and *MYB*, with more information on related genes available in Figure 7A. Due to the high explanatory power and lowest *p*-value and FDR values of *IER3*, it is considered the key “switch” gene in this analysis.

Further, SSCs were divided into *IER3*+ and *IER3*− groups based on IER3 expression levels, and cell interaction analysis explored the interactions between these gene expression levels and other cells. Results showed significantly enhanced interactions between IER3+ SSC and endothelial cells (Figure 7B). From the perspective of cell interaction signaling molecules (Figure 7C), many ligand–receptor signaling molecules were involved in the interactions between *IER3*+ SSC and endothelial cells, with MDK–NCL remaining a primary communication signal. Additionally, pathways such as MIF—(*CD74* + *CD44*) and MDK—(*ITGA6* + *ITGB1*) also displayed extensive intercellular interactions. These findings suggest that the IER3 gene plays a crucial regulatory role during cell development, particularly acting as a key “switch” gene in the early development of epithelial cells. The high expression of IER3 not only enhanced interactions between SSC and endothelial cells but also promoted intercellular communication through specific signaling molecules, such as MDK–NCL and MIF—(*CD74* + *CD44*). These findings provide new perspectives on the potential role of IER3 in CRC pathogenesis and suggest that this gene could be an important target for future research and therapy.

### 2.7. Impact of IER3 on CRC Epithelial Cells

To further clarify the impact of *IER3* on CRC epithelial cells, we assessed differential genes between *IER3*+ SSC and *IER3*− SSC (Figure 8A, Appendix A). The distribution of differential genes showed higher expression of the CXCL family genes (such as *CXCL2*, *CXCL3*, and *CXCL8*) in *IER3*+ SSC, while genes like *LMO7*, *SPATS2L*, *OPTN*, *DDX60L*, and *RNF213* were underexpressed. GSEA results also indicated a more active state in *IER3*+ SSC (Appendix A), with the top four activated gene sets including TNFα Signaling via NF-κB, Hypoxia, Inflammatory Response, and p53 Pathway (Figure 8B–E). Changes in metabolic pathways (Figure 8F), such as vitamin B6 metabolism, thiamine metabolism, nicotinate and nicotinamide metabolism, metabolism of xenobiotics by cytochrome P450, glycosphingolipid biosynthesis, and drug metabolism, were significantly enhanced in the *IER3* highly expressed SSC cell group. These findings suggest that *IER3* might be involved in the functional regulation of CRC epithelial cells through the control of CXCL family gene expression and the activity of metabolic pathways, potentially playing a key role in the onset and development of CRC. 

## 3. Discussion

This study employed scRNA-seq technology combined with MR analysis to uncover the causal relationship between characteristic genes of CRC-related serrated lesion cell subgroups and CRC pathogenesis. The results highlight that gene expressions in SSCs are closely linked to the onset of CRC, offering new insights into the heterogeneity of CRC and potential therapeutic targets. From single-cell sequencing data analysis to MR analysis outcomes, genes such as *RNASET2*, *ANXA4*, and *TMBIM1* exhibited a positive causal effect on CRC risk, whereas *RAB11A*, *IER3*, *SCARB2*, *ATF3*, and *CXCL8* demonstrated a negative causal effect. Notably, our findings reveal an apparent paradox regarding *IER3* function that merits careful consideration. While MR analysis, which utilizes GWAS data comparing cancer patients to healthy controls at the population level, suggests that *IER3* expression is associated with reduced CRC risk, our single-cell analysis demonstrates that *IER3*+ SSCs within established tumors exhibit enhanced pro-tumorigenic characteristics. This contradiction likely reflects the context-dependent and cell identity-specific roles of *IER3*. At the systemic level, *IER3* may exert protective effects against CRC initiation across diverse cell types and tissues. However, within the established tumor microenvironment, *IER3* upregulation in specific epithelial cell subpopulations (SSCs) may represent an adaptive response that enhances cellular survival and promotes tumor progression. This dual functionality underscores the complexity of gene function across different cellular contexts and disease stages.

Subsequent studies revealed a significant positive causal relationship for the *IER3* gene, suggesting its critical regulatory role in CRC development, particularly through modulating intercellular signaling within the tumor microenvironment. To further clarify the role of *IER3* in CRC, in-depth analysis based on single-cell data revealed enhanced interactions between *IER3*-positive SSCs and other cell types, notably with endothelial cells. The high correlation of *IER3* expression in SSCs suggests its pivotal regulatory role in CRC progression. Subsequent gene enrichment analyses confirmed *IER3′s* central role in regulating the cell cycle, immune response, and metabolic pathways, suggesting that *IER3* warrants further investigation as a candidate for therapeutic research. Additionally, cellular trajectory analysis affirmed the regulatory impact of *IER3* on intercellular interactions during epithelial cell development, reinforcing its crucial role in CRC progression. Overall, this study unveils the causal relationship between CRC serrated lesion characteristic cells and their associated genes with CRC risk, offering new perspectives on the pathogenesis and heterogeneity of CRC. These findings not only enrich our understanding of cell functions within the CRC microenvironment but also provide potential molecular targets for precision medicine. By further deciphering the relationships between these key genes and CRC, we anticipate identifying new therapeutic strategies that could enhance treatment efficacy and improve prognosis for CRC patients.

From a clinical perspective, this study may have potential clinical implications, particularly in the prevention, diagnosis, and treatment of CRC. Firstly, for early diagnosis and risk assessment, the study reveals a causal link between SSCs and their gene expressions with CRC risk. There are similarities to the results that have been published [29,30,31,32]. These expression patterns may provide insights for future diagnostic biomarker development and, pending validation in independent clinical cohorts, may contribute to risk assessment strategies before symptoms become apparent, thus allowing for early intervention. For instance, the detection of these characteristic genes in blood or bodily fluid samples can assess an individual’s risk of developing CRC. Secondly, in developing personalized treatment strategies, key genes like IER3 identified in this study have their roles in CRC progression further clarified, suggesting their potential as effective therapeutic targets [33,34,35]. Small-molecule drugs or monoclonal antibodies targeting these genes warrant investigation as potential therapeutic targets [36]. Moreover, personalized treatment plans can be tailored based on the expression of these genes in patients’ tumors, enhancing the specificity and efficacy of treatments. Thirdly, for monitoring treatment responses and recurrence, changes in gene expression can be used to assess treatment effectiveness and monitor for disease recurrence. By regularly tracking the expression levels of relevant genes in patients, physicians can timely understand treatment outcomes, adjust therapeutic strategies, and provide early warnings of potential relapses, offering more detailed post-treatment management [37,38]. These clinical applications demonstrate the potential contributions of this study across the entire management spectrum of CRC—from prevention and diagnosis to treatment—potentially transforming current clinical practices and improving patient survival rates and quality of life.

This study combines scRNA-seq with MR to reveal the causal relationship between specific types of epithelial cells in the colon and CRC, providing new insights into the cellular role transitions during CRC progression. By deeply analyzing the epithelial cell subpopulation within serrated lesions-SSCs-this research not only clarifies their contributions to CRC development but also uncovers potential molecular mechanisms, offering a scientific basis for future diagnostic and therapeutic strategies [39,40]. The distribution changes in this cell subpopulation not only reflect the heterogeneity of tumor tissue but may also indicate different biological behaviors of the tumor [6,41,42]. Additionally, the immunomodulatory properties of these cells, such as affecting lymphocyte-mediated immune responses, might enable the tumor to evade the host immune system through immune escape mechanisms. Moreover, gene expression analysis further reveals functional differences between these subgroups, such as SSCs possibly supporting tumor spread by activating cell growth-related signaling pathways [43,44,45], while a reduction in ABS cells might weaken their contribution to intestinal homeostasis [46,47]. These gene characteristics may also provide clues for developing new targeted therapies. In summary, the current analysis results at the molecular mechanism level can be briefly summarized in the following aspects: Firstly, the contribution of SSCs to CRC is affirmed. As an important epithelial cell subpopulation within the tumor microenvironment, SSCs play a key role in the development of CRC [48,49]. scRNA-seq technology allows us to detail the gene expression profile of SSCs, and MR analysis confirms the causal relationship between these expression patterns and CRC risk. Results suggest that SSCs play a significant role in promoting tumor proliferation and invasiveness, with their unique gene expression patterns involving many key biological processes such as cell adhesion, immune escape, and inflammatory responses [44,50,51]. For example, enhanced cadherin and cytokine activity in SSCs not only facilitate tight intercellular connections but may also promote tumor cell survival and spread by modulating the local immune environment. Additionally, these cells also play a significant role in the remodeling of the extracellular matrix, as evidenced by their high enrichment in focal adhesion and extracellular matrix pathways.

Secondly, the significance of genetic loci is addressed. By integrating public GWAS data and single-cell expression data, we identified several important genetic loci related to SSC cell expression. These loci show significant correlations with CRC risk in MR analysis. For instance, specific SNPs exhibit significant eQTL effects on key genes regulating cadherin and cytokine expression in SSCs. These genetic-level findings not only provide genetic evidence for the role of SSCs in CRC but also enhance their potential as therapeutic targets. Additionally, the discovery of these genetic loci may help identify more personalized treatment plans, optimizing treatment effects by targeting specific genetic variations.

Thirdly, the role and mechanisms of the *IER3* gene are explored. Known as an immediate early response gene, *IER3* plays important roles in various biological processes, especially in regulating cellular responses to external stimuli [52]. In this study, the expression of *IER3* in SSCs is closely related to the development of CRC, involving it in regulating the cell cycle, cell proliferation, and apoptosis through pathways like FOS proteins and *NF-κB.* MR analysis shows that changes in *IER3* expression correlate with CRC risk, highlighting its potential role in tumor biology. Further cellular functional assays and signaling pathway analyses validate that upregulation of *IER3* significantly enhances the proliferative capabilities of CRC cells, potentially promoting inflammatory responses and cell survival signals through the *NF-κB* pathway [33,53,54,55]. Additionally, *IER3* may also regulate cell proliferation and apoptosis by affecting the regulation of key cell cycle proteins such as cyclins and CDK inhibitors [56]. In conclusion, this study, through high-level integrated analysis, reveals the molecular characteristics and functions of SSCs in CRC, as well as the mechanisms of action of key genes such as *IER3*, providing important scientific evidence for future precision diagnosis and treatment targeting CRC. These findings not only enhance our understanding of the complex pathological processes of CRC but also identify new therapeutic targets, potentially advancing the development of personalized medical strategies, thereby improving clinical treatment outcomes and quality of life for CRC patients.

Although this study provides important insights into the causal relationship between specific epithelial cell subpopulations and CRC risk, several limitations must be acknowledged. First, data and technical limitations. The absence of eQTL data for some differentially expressed genes may lead to omission of key regulatory factors in our MR analysis. Our reliance on public single-cell databases with limited patient demographic and clinical information (age, tumor stage, molecular subtypes, treatment history) constrains generalizability across diverse patient populations and clinical contexts. Importantly, we did not apply imputation methods (such as MAGIC) to address the inherent sparsity and dropout effects in single-cell RNA-seq data, which may result in underdetection of biologically relevant but weakly expressed markers, particularly in rare cell populations like SSCs. Technical variability in sequencing, including batch effects and differential cell capture efficiency, could also affect analytical accuracy. Second, MR analytical constraints. Despite sensitivity analyses, our MR approach depends on strict instrumental variable assumptions that may be violated through genetic heterogeneity or pleiotropy, potentially introducing residual bias. Third, pathway and immune landscape gaps. Our analyses did not reveal significant enrichment of canonical pathways like Wnt signaling, despite their established roles in serrated CRC. We also did not comprehensively assess immune checkpoint ligand expression or SSC interactions with diverse immune populations, limiting understanding of potential immune evasion mechanisms. Fourth, experimental validation. Our observational study design lacks functional validation through laboratory experiments. The apparent contradiction regarding *IER3* function (protective in MR analysis versus pro-tumorigenic in single-cell data) exemplifies the need for comprehensive functional studies to resolve mechanistic complexities [57]. Future research should incorporate imputation methods for single-cell data processing, gene knockout/overexpression studies, independent clinical validation cohorts, and targeted pathway investigations to confirm the biological significance and therapeutic relevance of these findings. In conclusion, while this study provides valuable insights into CRC pathogenesis, these limitations emphasize the need for cautious interpretation and comprehensive methodological improvements in future investigations.

## 4. Materials and Methods

### 4.1. Single-Cell RNA Sequencing Data Retrieval and Processing for CRC

CRC-related scRNA-seq data were retrieved from the Gene Expression Omnibus (GEO) database [58]. Quality control was performed using the ‘Seurat’ R package (version 4.4.0) [59] to select high-quality cells and eliminate low-quality cells and potential doublets. The filtering criteria were 200–4000 genes detected per cell, >100 features per cell, and mitochondrial gene percentage ≤50%. Subsequent analyses included principal component analysis (PCA) for dimensionality reduction, followed by visualization using Uniform Manifold Approximation and Projection (UMAP). Furthermore, preliminary annotations of cell clusters were performed using known marker genes (B cells: *MS4A1*, *CD79A*; endothelial cells: *PECAM1*, *CDH5*; epithelial cells: *EPCAM*, *KRT8*; monocytes: *CD14*, *LYZ*; NK cells: *KLRD1*, *NKG7*; T cells: *CD3D*, *CD2*; smooth muscle cells: *ACTA2*, *MYH11*, *TAGLN*; tissue stem cells: *ASCL2*, *OLFM4*), allowing for the exploration of potential differences and biological processes among cell groups.

### 4.2. Identification and Functional Enrichment Analysis of CRC-Related Serrated Characteristic Cells

After initial cell type annotations, subpopulations identified as epithelial cells were selected and verified using known epithelial cell markers, including *EPCAM*, *KRT8*, and *KRT18* [60,61]. Subsequently, these epithelial subpopulations underwent normalization and high-variable gene selection, followed by re-clustering. UMAP was utilized to visualize the clustering results, illustrating the distribution of epithelial subpopulations. Differential expression analysis was performed to identify marker genes specific to each subpopulation. Via the ‘clusterProfiler’ R package [62], Gene Ontology (GO) and Kyoto Encyclopedia of Genes and Genomes (KEGG) enrichment analyses were conducted to elucidate the functional characteristics of each subpopulation [63,64]. To infer cell differentiation trajectories, pseudotime analysis tools such as the ‘Monocle 2’ R package (version 2.32.0) [65,66]. These analyses allowed us to analyze the positioning of serrated characteristic cells within these trajectories. Furthermore, gene regulatory networks for serrated cells were constructed to identify key transcription factors and signaling pathways. Comparative analyses were conducted between serrated characteristic cells and other epithelial subgroups.

### 4.3. Expression and Functional Analysis of Core Genes at the Single-Cell Level in CRC

Differential expression analysis between SSCs and non-SSCs was conducted using the entire scRNA-seq dataset to identify differentially expressed genes (DEGs). The genes identified in both analyses were considered candidate marker genes for SSCs and were used as exposure variables in subsequent MR analyses. We constructed the protein-protein interaction (PPI) network for DEGs using the ‘STRING’ database [67]. Functional enrichment analysis of core genes was performed using the ‘clusterProfiler’ R package, highlighting their significant roles in regulating the cell cycle, signaling pathways, metabolism, and immune responses [68]. This analysis revealed substantial enrichment in biological processes, molecular functions, and cellular components and further explored the critical roles of these genes in disease onset and progression. Combined with KEGG analysis, this evidence revealed both direct and indirect connections between genes and identified several key nodes that play central roles in critical biological processes such as signaling, cell proliferation, and apoptosis. Furthermore, enrichment analysis enabled the identification of potential functional and biological pathways, providing a scientific basis for further mechanistic studies and potential drug target development.

### 4.4. MR Analysis of SSCs Marker Genes and CRC Expression Quantitative Trait Loci (eQTLs)

Single-nucleotide polymorphisms (SNPs) associated with candidate marker genes were extracted from eQTLGen 2019 data (primarily blood-derived) in the OpenGWAS database [69,70], using CRC data from the FinnGen database as outcomes [71]. Two-sample MR analysis was employed to explore the causal relationships between genes and CRC. SNP selection criteria included a *p*-value threshold of <5 × 10^−8^, r^2^ = 0.001, and a distance of 10,000 kb [72,73]. The robustness of the instrumental variables (IVs) included in the analysis was measured by calculating the F-statistic (F = β^2^/SE^2^, F > 10) [74,75]. The main method for MR analysis was inverse-variance weighted (IVW), supplemented by MR-Egger, weighted median, and weighted mode methods [76,77]. The significance of the results was determined by the consistency in the direction of causal effect estimates across all methods (either all greater than zero or all less than zero) and required a *p*-value < 0.05 in the IVW analysis [78]. Sensitivity analyses, including leave-one-out analysis, single SNP causal estimation, and assessments of heterogeneity (Cochran’s Q test, where *p* < 0.05 indicates the presence of heterogeneity) and pleiotropy (MR-Egger intercept test, where *p* < 0.05 indicates the presence of pleiotropy), were performed to ensure the robustness and reliability of the causal inference results [79]. Genes demonstrating significant causal associations in the MR analysis were designated as core genes for subsequent investigations.

### 4.5. GeneSwitches Analysis of Core Genes

Using previously obtained scRNA-seq data from CRC, the expression trends of the identified core genes across different cell types were reassessed. Based on trajectory analysis results for epithelial cells in CRC single-cell data, the ‘GeneSwitches’ R package (version 0.1.0) was used to perform binary analysis on genes during the epithelial cell differentiation trajectory [80]. This analysis identified potential switch genes exhibiting on and off expression states. Logistic regression and McFadden’s Pseudo R^2^ pseudotime correlation analyses were conducted on these potential switch genes. For each switch gene, the switching time point was estimated through logistic regression, and the associated R^2^ value, obtained from pseudotime correlation analysis, indicated whether the gene’s expression activation correlated positively with pseudotime (R^2^ > 0, defined as upregulation) or negatively (R^2^ < 0, defined as downregulation). A higher pseudotime correlation suggests a closer relationship between the gene and the differentiation trajectory. After determining each potential switch gene’s switching time and associated R^2^ value, the top switch genes were visually displayed in order of their switching times along the differentiation trajectory, providing clearer insights into the roles of key genes during the process.

### 4.6. Impact of Key Switch Genes on CRC Cell Interactions and Biological Processes

Tumor cells in the CRC scRNA-seq dataset were reclassified into high and low expression groups based on the expression levels of significant switch genes. Cell interaction analysis methods were employed to evaluate the impact of varying expression levels of these switch genes on other cell types, revealing how these gene expression changes regulate intercellular signaling and cell functions within the microenvironment. Additionally, Gene Set Enrichment Analysis (GSEA) was used to assess differences in gene set activation and suppression between high- and low-expression cell groups [81], further clarifying the impact of gene expression levels on colorectal cancer epithelial cells. This analysis also explored changes in metabolic pathways between the two cell groups, providing new insights into tumor heterogeneity and its impact on the microenvironment, thereby offering theoretical support for precision treatment strategies targeting these key genes.

## 5. Conclusions

This study demonstrates that specific serrated epithelial cell subpopulations in CRC are closely associated with the disease’s progression, particularly through the actions of key genes such as *IER3*, showing a potential causal link. Further research is needed to determine whether these cell subpopulations can serve as effective markers for risk prediction to guide therapeutic interventions, clarify any expression–response relationships, and describe the nature of this association across population subgroups.

## Figures and Tables

**Figure 1 ijms-26-07187-f001:**
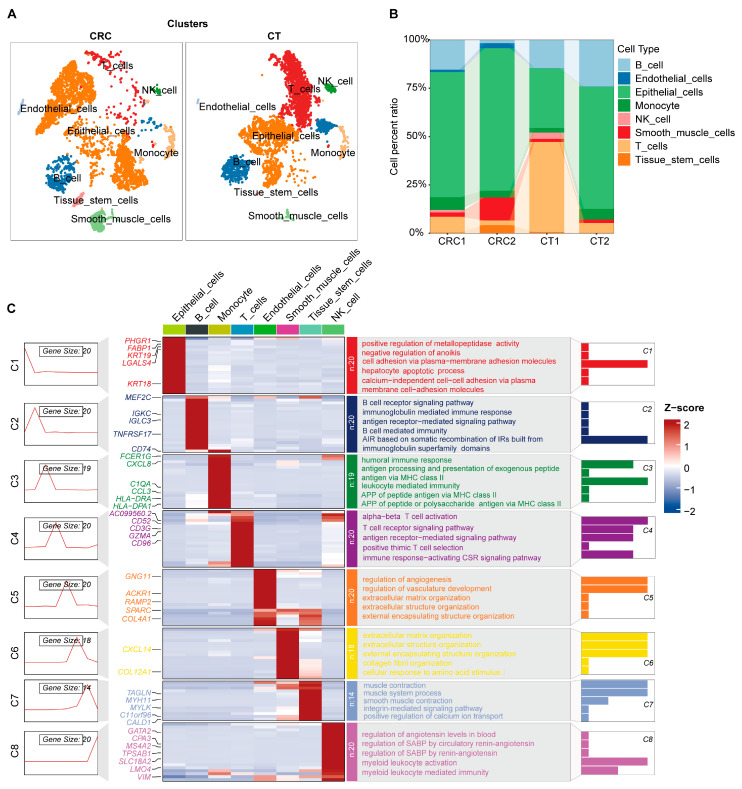
Single cell atlas (**A**) and cell proportion (**B**) of colorectal cancer (CRC), representative marker gene expression, and biological functions of different cell groups (**C**). CRC, colorectal cancer; CT, healthy control group. AIR, adaptive immune response; IRs, immune receptors; APP, antigen processing and presentation; CSR, cell surface receptor; SABP, systemic arterial blood pressure.

**Figure 2 ijms-26-07187-f002:**
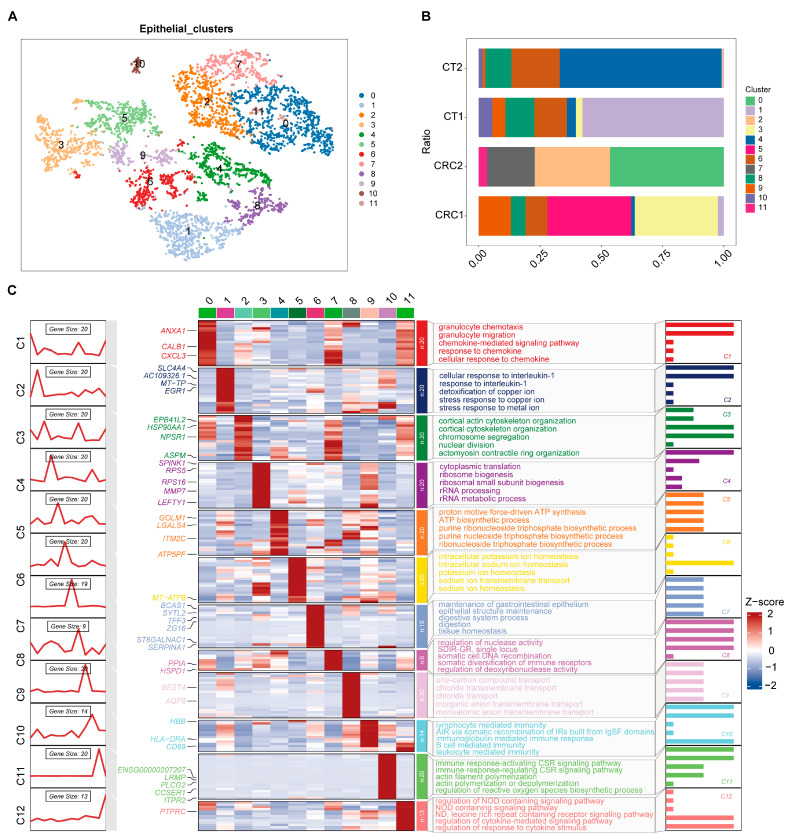
Cell clusters atlas (**A**) and cell proportion (**B**) of epithelial cells from the single-cell RNA sequence data of colorectal cancer (CRC), representative marker gene expression, and biological functions of different cell clusters (**C**). CRC, colorectal cancer; CT, healthy control group; NOD, nucleotide-binding oligomerization domain; ND, nucleotide-binding domain; IgSF, immunoglobulin superfamily; AIR, adaptive immune response; IRs, immune receptors; CSR, cell surface receptor.

**Figure 3 ijms-26-07187-f003:**
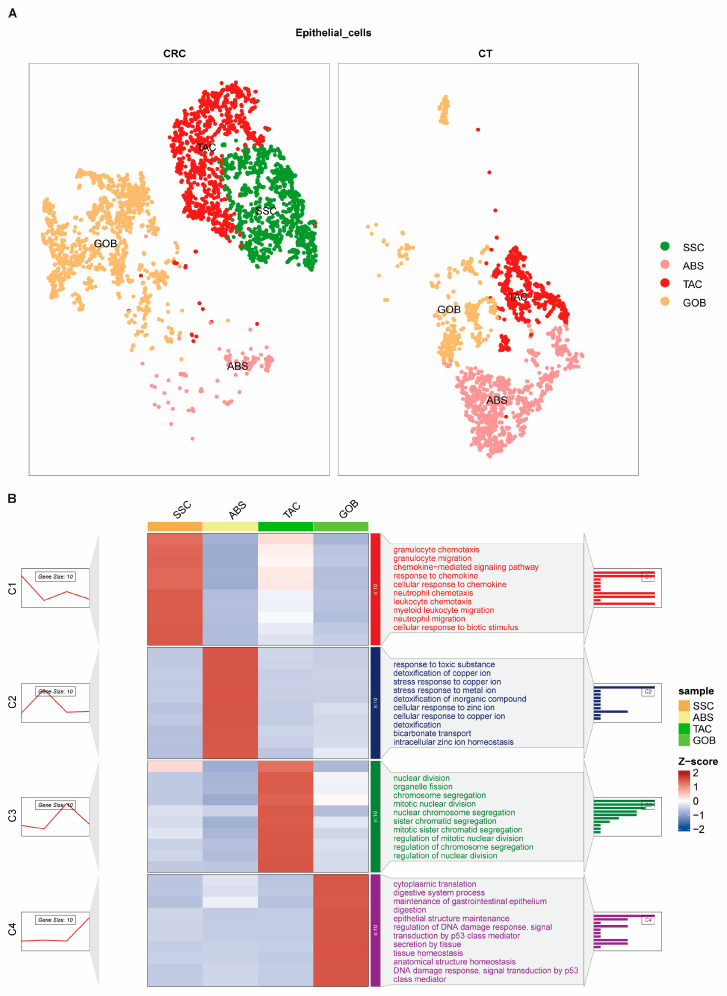
The atlas (**A**) of four subtypes of epithelial cells for colorectal cancer (CRC) and representative marker gene expression and biological functions of different epithelial cells (**B**). CRC, colorectal cancer; CT, healthy control group; SSC, specific serrated cells; ABS, absorptive cells; TAC, transit amplifying cells; GOB, goblet cells.

**Figure 4 ijms-26-07187-f004:**
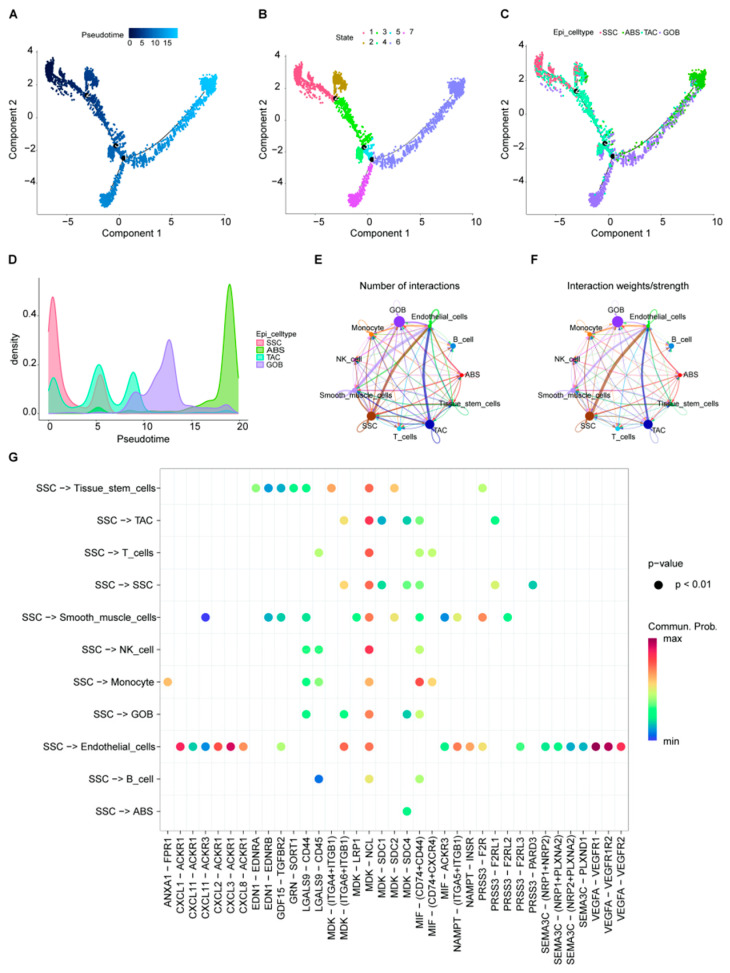
Trajectory analysis of four subgroups of epithelial cells and their interaction with other cells from colorectal cancer (CRC). (**A**), pseudotime analysis of epithelial cell development, with dark blue representing the early stage and light blue representing the late stage. (**B**), classify the developmental status of epithelial cells based on trajectory analysis of developmental nodes. (**C**), the distribution of four different subgroups of epithelial cells along the developmental trajectory. (**D**), the distribution of cell density along the developmental trajectory of four different subgroups of epithelial cells. (**E**), comparison of the number of interactions between four different subgroups of epithelial cells and other cells in colorectal cancer. (**F**), comparison of the intensity of interaction between four different subgroups of epithelial cells and other cells in colorectal cancer. (**G**), the distribution characteristics of ligand-receptor interactions between the SSC epithelial cell subgroup and all cells. SSC, specific serrated cells; ABS, absorptive cells; TAC, transit amplifying cells; GOB, goblet cells.

**Figure 5 ijms-26-07187-f005:**
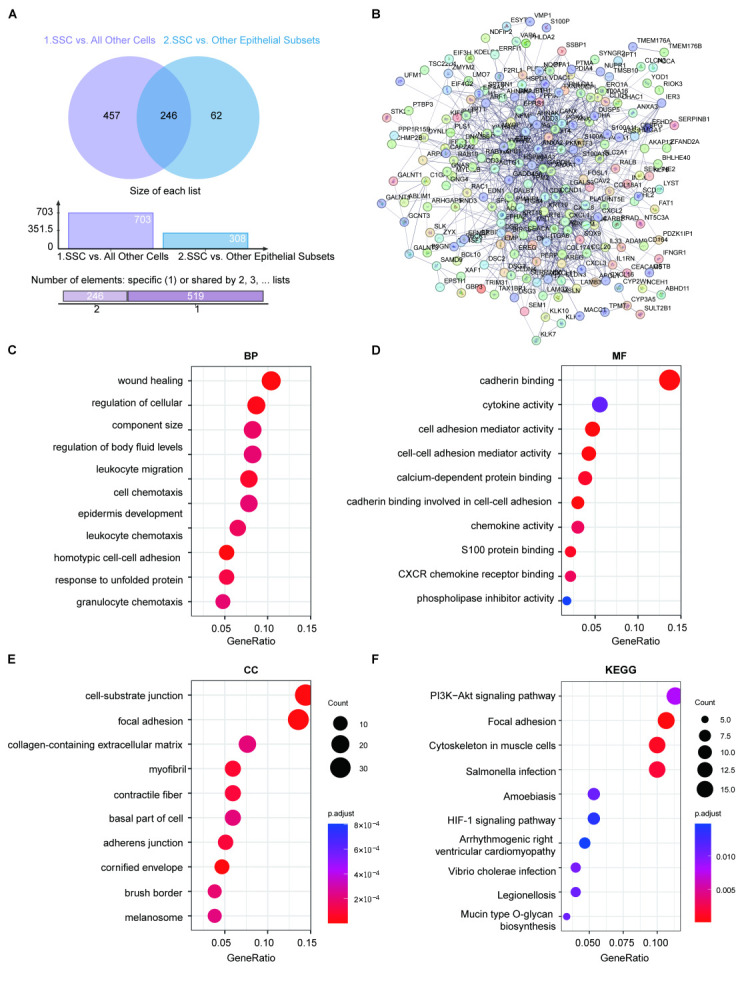
Differential gene analysis and enrichment analysis of specific serrated cells (SSCs). (**A**), the results of differential gene analysis between SSC cells and all colorectal cancer (CRC) cells, as well as differential analysis with non-SSC epithelial cells. (**B**), a protein interaction network consisting of 246 differentially expressed genes. (**C**), enrichment analysis of biological processes (BP) of differentially expressed genes. (**D**), enrichment analysis of molecular functions (MF) of differentially expressed genes. (**E**), enrichment analysis of molecular components (CC) of differentially expressed genes. (**F**), enrichment analysis of KEGG signaling pathways of differentially expressed genes.

**Figure 6 ijms-26-07187-f006:**
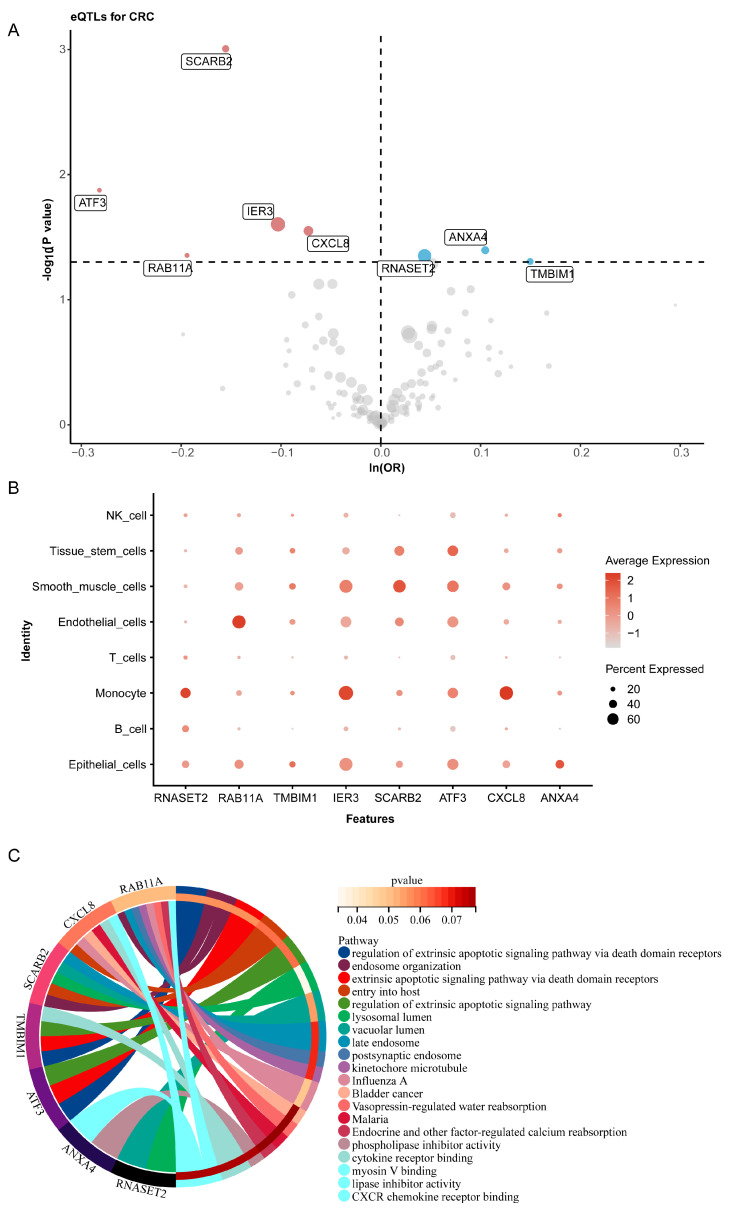
Mendelian randomization (MR) analysis was used to screen the causal relationship between the expression quantitative trait loci (eQTL) of differentially expressed genes and colorectal cancer (CRC), as well as to analyze the expression levels and functional enrichment of candidate genes. (**A**), the volcano plot shows the results of MR analysis between the differential gene eQTL related to specific serrated cells (SSCs) and CRC, where red and blue represent genes with significant MR analysis, and gray represents genes without significant MR analysis. (**B**) The bubble plot displays the expression levels of significant genes analyzed by MR in different cell groups of CRC single-cell data. (**C**). Functional enrichment analysis of significant genes in MR analysis.

**Figure 7 ijms-26-07187-f007:**
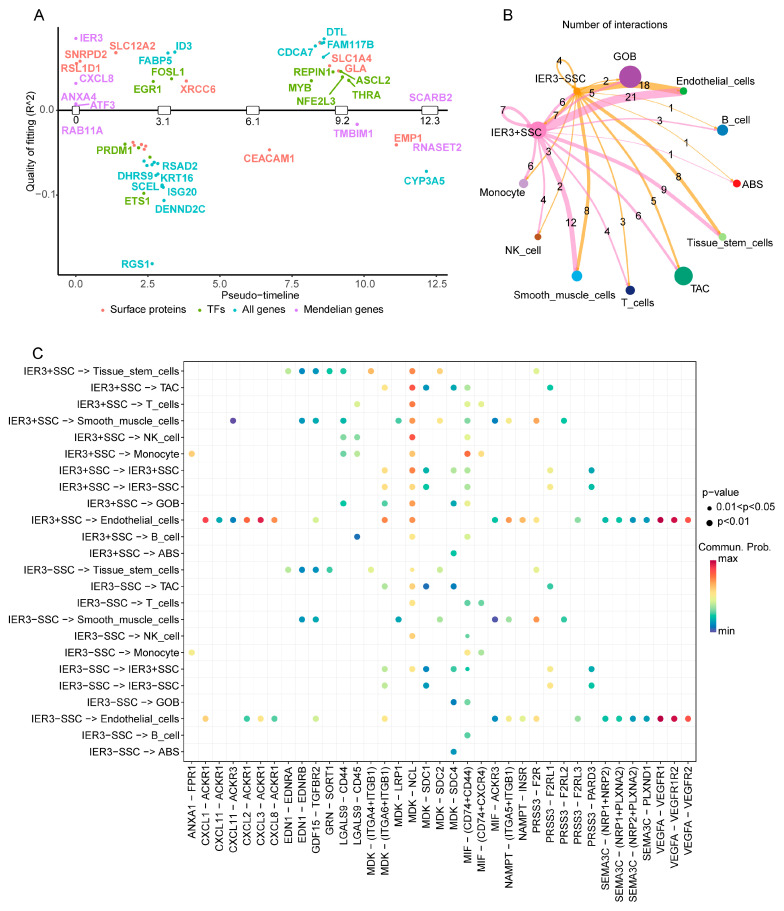
Switchgenes analysis for differential genes and cell chat analysis between specific serrated cells (SSCs) and other cells based on high and low expression of switch genes. (**A**), analysis of switch gene categories based on pseudotime from trajectory analysis. (**B**), cell chat number between SSCs with high and low expression of IER3 and other colorectal cancer (CRC) cells. (**C**), the distribution characteristics of ligand-receptor interactions between SSCs with high and low expression of IER3 and other CRC cells. TF, transcription factor; SSC, specific serrated cells; ABS, absorptive cells; TAC, transit amplifying cells; GOB, goblet cells.

**Figure 8 ijms-26-07187-f008:**
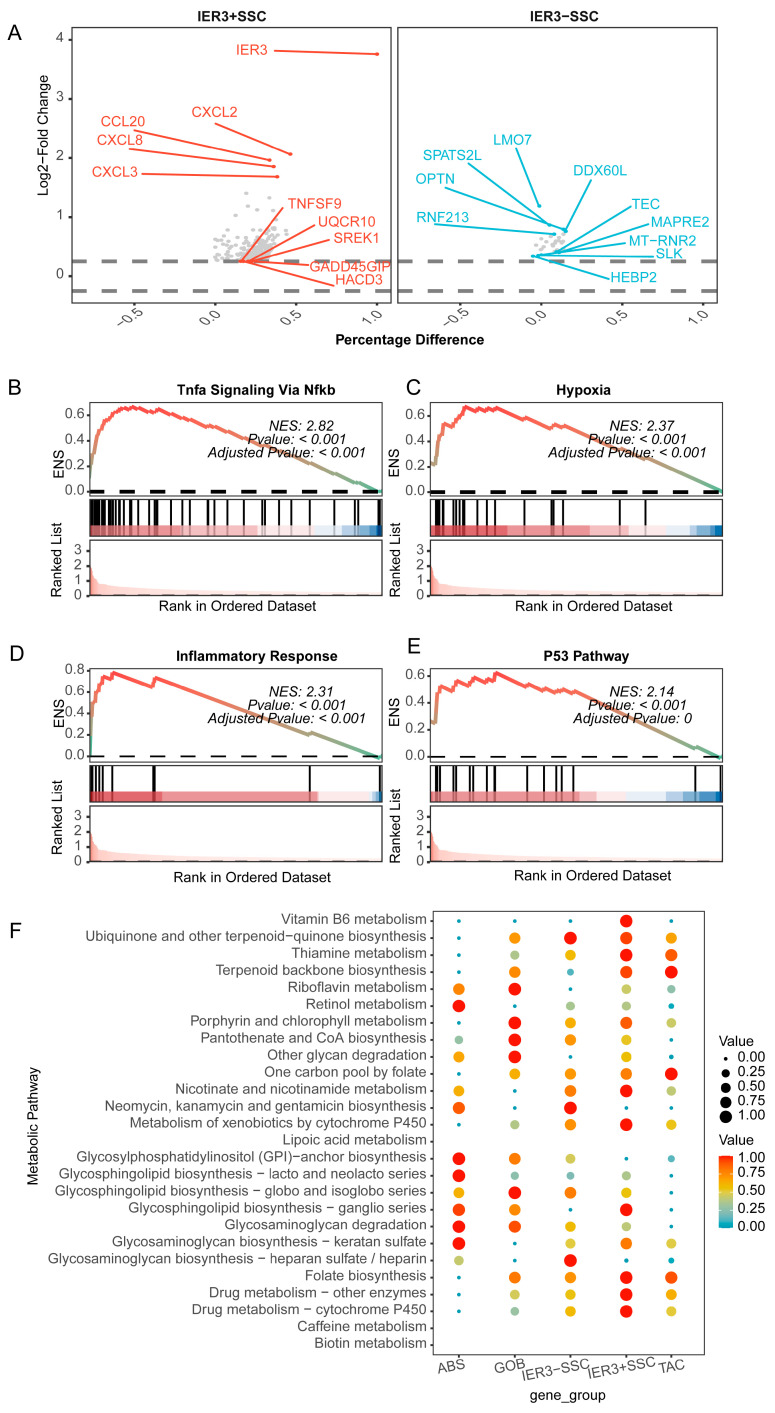
Differential analysis (**A**) and representative GSEA results (**B**–**E**) of specific serrated cells (SSCs) between high and low expression groups of IER3, as well as the regulation of metabolic pathways (**F**) by IER3 high and low expression. ENS, enrichment score; SSC, specific serrated cells; ABS, absorptive cells; TAC, transit amplifing cells; GOB, goblet cells.

**Table 1 ijms-26-07187-t001:** The results of GeneSwitches analysis for epithelial cells in colorectal cancer. Genes, The names of the genes analyzed; Time, refers to the time point at which the gene expression was measured based on monocle analysis; CI (Confidence Interval), a range of values that’s likely to contain the true value of the estimate; FDR (false discovery rate), the expected proportion of false discoveries among the rejected hypotheses; R^2^, the proportion of the variance for a dependent variable that’s explained by an independent variable in a regression model. Estimates the effect size or magnitude of the association. Direction, whether the gene expression is upregulated (“up”) or downregulated (“down”).

Genes	Time	CI	*p*-Values	FDR	R2	Estimates	Direction
*IER3*	0.00	1.75	3.89 × 10^−10^	4.85 × 10^−8^	8.48 × 10^−2^	0.32	up
*CXCL8*	0.00	1.93	3.31 × 10^−9^	2.66 × 10^−7^	3.18 × 10^−2^	0.17	up
*ANXA4*	0.00	5.92	2.62 × 10^−3^	2.00 × 10^−2^	7.71 × 10^−3^	0.08	up
*ATF3*	0.00	19.81	3.65 × 10^−2^	1.43 × 10^−1^	5.57 × 10^−3^	0.07	up
*RAB11A*	0.00	26.42	9.22 × 10^−2^	2.66 × 10^−1^	2.75 × 10^−3^	0.05	up
*SCARB2*	12.28	946.42	7.95 × 10^−1^	9.02 × 10^−1^	5.34 × 10^−5^	−0.01	down
*RNASET2*	12.28	59.52	2.60 × 10^−1^	4.96 × 10^−1^	1.01 × 10^−3^	−0.03	down
*TMBIM1*	9.76	4.90	3.00 × 10^−6^	8.16 × 10^−5^	1.65 × 10^−2^	−0.12	down

## Data Availability

The datasets generated and/or analysed during the current study are available in the ieu open GWAS project (https://gwas.mrcieu.ac.uk/, accessed on 5 March 2025) and GEO database (GSE221575, https://www.ncbi.nlm.nih.gov/gds, accessed on 5 March 2025). All analysis scripts and code used in this study will be provided upon reasonable request from corresponding authors.

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
