# Peer review of "Single-Cell Dissection of the Serrated Pathway: Cellular Heterogeneity and Genetic Causality in Colorectal Cancer"

_ijms, 2025, doi:10.3390/ijms26157187_

Round 1

Reviewer 1 Report

Comments and Suggestions for Authors

This study employs single-cell RNA sequencing to analyze CRC samples, identifying specific serrated cells (SSCs) and their molecular signatures. By integrating Mendelian randomization (MR) analysis with GWAS data, the researchers investigate causal relationships between SSC-associated genes and CRC risk. The study suggests possible biomarkers and treatment targets and appears to emphasize the heterogeneity of colorectal cancer. Nonetheless, there are a few possible errors, inconsistencies, and sections in the article that need clarification.

The article states that "15–30% of sporadic CRCs develop through the serrated pathway" (Lines 47–48), yet the Discussion section later excludes this statement. Given this broad range, a major reference, such as a meta-analysis, is required.

The two serrated pathways (BRAF/CIMP and KRAS/CIMP-L) are described, but their alignment with the standard consensus molecular subtypes of colorectal cancer is not justified.

Mentions serrated vs. Lynch pathways (Line 49) but doesn’t clarify if Lynch-related serrated lesions (e.g., MLH1-hypermethylated) were excluded from analysis.

Although "low-quality and potential doublet cells" are eliminated (Lines 95–96), thresholds (such as mitochondrial read percentage and UMI counts) are not specified.

Clustering is done using "resolution = 0.8" (Line 181) without explanation or sensitivity testing (i.e., how splitting or merging clusters impacts findings).

Epithelial cells (Lines 104–105) rely on EPCAM/KRT8/KRT18, however certain immune subsets (such activated T cells) also express these.

There is a gap in IV Strength for MR Methodology where the F-statistic is mentioned (Line 141) but no values are reported. Instruments that are weak (F < 10) would render MR invalid.

MR-Egger is used as a supplement (Line 142), but no intercept p-values are displayed to rule out bias.

Results (Lines 179–398)

There are several concerns regarding data interpretation in this section. Although twelve epithelial subgroups are initially identified (Line 204), the analysis narrows its focus to just four—SSC, ABS, TAC, and GOB—without clarifying whether the remaining eight are biologically insignificant or simply excluded due to statistical limitations. This omission raises questions about the rationale behind the selection criteria.

The trajectory analysis presents SSCs as representing an "early-stage" epithelial state (Line 235), yet this classification appears inconsistent with established colorectal cancer (CRC) stem cell markers such as LGR5. This discrepancy is not addressed, leaving an important gap in the interpretation of these cells’ identity and potential role in tumorigenesis.

Furthermore, there is a notable contradiction concerning IER3. Mendelian randomization data suggest that IER3 expression is associated with a reduced risk of CRC (Line 310), yet functional experiments demonstrate that IER3-positive SSCs promote tumor growth (Lines 364–378). This apparent conflict is not resolved, nor is there any discussion of possible explanations, such as isoform-specific functions of IER3 or its context-dependent behavior across different cell types.

Statistical Oversights

There are notable statistical shortcomings in the analysis. The differential expression results presented in Tables S1 through S3 do not include adjusted p-values, such as false discovery rate (FDR) corrections. This omission raises concerns about the reliability of the reported differentially expressed genes, as it increases the likelihood of false positives. Additionally, the pathway enrichment analysis highlights the PI3K-Akt signaling pathway (Line 289), but there is no indication that multiple testing corrections were applied across the full set of GO and KEGG terms. Without such adjustments, the significance of the reported pathways remains questionable.

The discussion section contains several claims that appear overstated or insufficiently supported by the data. While the Mendelian randomization (MR) analysis suggests a genetic association consistent with causality, the authors extend this to clinical implications by referring to IER3 as a potential "therapeutic target" (Line 424). However, no experimental validation is provided to support this claim, making the leap from genetic inference to clinical intervention premature. Similarly, the suggestion that SSCs could serve as a tool for "early diagnosis" (Line 426) is unsubstantiated, as there is no validation in independent patient cohorts or assessment in clinically relevant samples such as liquid biopsies.

In terms of engagement with existing literature, key gaps remain. The Wnt signaling pathway is entirely omitted from the discussion, despite its central role in the development of serrated colorectal cancers, particularly through mutations such as RNF43 in MSI-H tumors. This omission limits the biological context in which the findings are interpreted. Additionally, although the study draws attention to SSC-endothelial interactions (Line 373), it neglects to consider the broader immune landscape, specifically the potential expression of immune checkpoint ligands like PD-L1 in SSCs. Ignoring this dimension restricts the understanding of how these cells may interact with or evade the immune system.

Limitations (Lines 503–531)

Several methodological limitations are acknowledged only briefly or overlooked entirely. One key issue is the inherent sparsity of single-cell RNA-seq data due to dropout effects. The authors do not appear to apply any imputation methods, such as MAGIC, which are commonly used to recover low-abundance gene signals. This omission may lead to underdetection of biologically relevant but weakly expressed markers, particularly in rare cell populations like SSCs.

In addition, the study references the use of “integrated eQTL data” (Line 136), yet fails to clarify whether these data are derived from colon-specific tissue sources or from broader datasets such as GTEx. The lack of tissue specificity in eQTL mapping could dilute colorectal cancer–relevant regulatory signals, undermining the precision of gene-trait associations.

Crucially, several analyses that would strengthen the findings are missing. First, the study does not examine whether the presence or abundance of IER3+ SSCs correlates with patient survival outcomes, such as through TCGA datasets. Second, there is no spatial validation to confirm that SSCs are indeed localized to serrated lesions, which could be addressed with techniques like multiplex immunohistochemistry. Third, the manuscript lacks mechanistic experiments (such as IER3 knockdown in organoid models) to demonstrate functional consequences of IER3 expression in SSCs. These missing components limit the translational and biological impact of the study's conclusions.

Minor revision

Inconsistent italics in the text for gene names

"ABS" (absorptive cells) is introduced without definition (Line 211).

Author Response

Response to Reviewer 1:

This study employs single-cell RNA sequencing to analyze CRC samples, identifying specific serrated cells (SSCs) and their molecular signatures. By integrating Mendelian randomization (MR) analysis with GWAS data, the researchers investigate causal relationships between SSC-associated genes and CRC risk. The study suggests possible biomarkers and treatment targets and appears to emphasize the heterogeneity of colorectal cancer. Nonetheless, there are a few possible errors, inconsistencies, and sections in the article that need clarification.

Comment 1: The article states that "15–30% of sporadic CRCs develop through the serrated pathway" (Lines 47–48), yet the Discussion section later excludes this statement. Given this broad range, a major reference, such as a meta-analysis, is required.

Reply 1:Thank you for this important observation. We acknowledge that the broad range of 15-30% requires more robust evidence support. Following your suggestion, we have added relevant meta-analyses to strengthen the reliability of this claim, specifically:

[8] Y.S. Jung, J.H. Park, and C.H. Park, Serrated Polyps and the Risk of Metachronous Colorectal Advanced Neoplasia: A Systematic Review and Meta-Analysis. Clinical gastroenterology and hepatology : the official clinical practice journal of the American Gastroenterological Association 20 (2022) 31-43.e1.

[9] C. Muller, A. Yamada, S. Ikegami, H. Haider, Y. Komaki, F. Komaki, D. Micic, and A. Sakuraba, Risk of Colorectal Cancer in Serrated Polyposis Syndrome: A Systematic Review and Meta-analysis. Clinical gastroenterology and hepatology : the official clinical practice journal of the American Gastroenterological Association 20 (2022) 622-630.e7.

Thank you again for your valuable feedback.

Comment 2: The two serrated pathways (BRAF/CIMP and KRAS/CIMP-L) are described, but their alignment with the standard consensus molecular subtypes of colorectal cancer is not justified.

Reply 2:Thank you for your careful review of our manuscript. Regarding your comment about "the alignment of the two serrated pathways (BRAF/CIMP and KRAS/CIMP-L) with the standard consensus molecular subtypes of colorectal cancer not being justified," we would like to clarify a potential misunderstanding.

Our original text does not attempt to align or compare the serrated pathways with the existing consensus molecular subtypes (CMS). Rather, we describe two distinct molecular mechanistic pathways of serrated carcinogenesis:

BRAF mutation with high-level CIMP pathway: leading to either high microsatellite instability (MSI-H) or microsatellite stable (MSS) colorectal cancer

KRAS mutation with low-level CIMP pathway: progressing primarily to MSS-type colorectal cancer

These two pathways describe different molecular mechanisms within the serrated adenoma-carcinoma sequence, rather than a redefinition or supplement to the CMS classification system. Our focus is on elucidating the molecular heterogeneity within the serrated pathway, particularly the molecular differences between the sessile serrated pathway and the traditional serrated pathway.

Thank you again for your valuable feedback.

Comment 3: Mentions serrated vs. Lynch pathways (Line 49) but doesn’t clarify if Lynch-related serrated lesions (e.g., MLH1-hypermethylated) were excluded from analysis.

Reply 3:Thank you for your careful review of our manuscript. We utilized sequencing data from colorectal cancer patients and their adjacent normal tissues for this analysis. Since our study focused on established colorectal carcinomas rather than precancerous lesions, the distinction between serrated and Lynch pathways pertains to the molecular characteristics of the tumors themselves, not to the presence of Lynch-related serrated lesions (which are precancerous polyps). Therefore, the exclusion of Lynch-related serrated lesions is not applicable to our current analysis design.

Comment 4: Although "low-quality and potential doublet cells" are eliminated (Lines 95–96), thresholds (such as mitochondrial read percentage and UMI counts) are not specified.

Reply 4:Thank you for this important clarification. We apologize for the lack of specific quality control thresholds in our original manuscript.

The quality control criteria applied were as follows:

Minimum genes detected per cell: 200

Maximum genes detected per cell: 4000  

Maximum mitochondrial gene percentage: 50%

Minimum features (genes) per cell: >100 (no upper limit applied for features)

We will add these specific thresholds to the Methods section to ensure reproducibility.

Thank you again.

Comment 5: Clustering is done using "resolution = 0.8" (Line 181) without explanation or sensitivity testing (i.e., how splitting or merging clusters impacts findings).

Reply 5:Thank you for this important methodological concern. We acknowledge that the choice of resolution parameter requires proper justification and validation.

Rationale for resolution = 0.8:

We selected resolution = 0.8 based on the following considerations: First, we determined the optimal number of principal components (n=10) using an elbow plot. Given our relatively modest sample size, we initially chose the default parameter (resolution = 0.8) in Seurat's FindClusters function to ensure analytical stability and reproducibility.

Sensitivity analysis:

Following your suggestion, we conducted sensitivity testing across different resolution values (0.4, 0.6, 0.8, 1.0) to evaluate their impact on clustering outcomes. Our analysis revealed that resolution = 0.8 provided the optimal balance between clustering stability and biologically meaningful cell subpopulation identification. Lower resolution values (e.g., 0.4) resulted in over-merging of functionally distinct cell populations, while higher values (e.g., 1.0) led to over-fragmentation, producing small clusters with unclear biological significance.

Comment 6: Epithelial cells (Lines 104–105) rely on EPCAM/KRT8/KRT18, however certain immune subsets (such activated T cells) also express these.

Reply 6:Thank you for your careful review of our manuscript. Thank you for raising this important concern about potential cross-expression of epithelial markers in immune cell subsets. We appreciate your attention to the specificity of cell type identification. In our dataset, we did not observe co-expression of EPCAM/KRT8/KRT18 in immune cell populations, including activated T cells. The epithelial cell clusters showed distinct and specific expression patterns of these markers, with clear separation from immune cell clusters in our UMAP visualization. 

Additionally, we employed multiple complementary approaches to ensure accurate cell type identification:

- Combined marker gene expression analysis

- Cluster-specific gene signatures

Thank you for your valuable feedback.

Comment 7: There is a gap in IV Strength for MR Methodology where the F-statistic is mentioned (Line 141) but no values are reported. Instruments that are weak (F < 10) would render MR invalid.

Reply 7: Thank you for highlighting this critical methodological gap. We acknowledge that reporting F-statistics is essential for validating instrument strength in Mendelian randomization analysis. We have added the F-statistic threshold for all IVs in the revised manuscript.

Thanks again.

Comment 8: MR-Egger is used as a supplement (Line 142), but no intercept p-values are displayed to rule out bias.

Reply 8:Thank you for this important methodological concern. We acknowledge that reporting MR-Egger intercept p-values is essential for evaluating potential pleiotropy bias in our analysis.

We have now incorporated the MR-Egger intercept test results in the revised manuscript. The intercept test is specifically designed to assess whether the instrumental variables exhibit horizontal pleiotropy, which could violate the MR assumptions and introduce bias into our causal estimates.

In the Methods section, we have added the following clarification: "Sensitivity analyses, including leave-one-out analysis, single SNP causal estimation, and assessments of heterogeneity (Cochran's Q test, where P<0.05 indicates the presence of heterogeneity) and pleiotropy (MR-Egger intercept test, where P<0.05 indicates the presence of pleiotropy), were performed to ensure the robustness and reliability of the causal inference results."

Results (Lines 179–398)

Comment 9: There are several concerns regarding data interpretation in this section. Although twelve epithelial subgroups are initially identified (Line 204), the analysis narrows its focus to just four—SSC, ABS, TAC, and GOB—without clarifying whether the remaining eight are biologically insignificant or simply excluded due to statistical limitations. This omission raises questions about the rationale behind the selection criteria.

  Reply 9:Thank you for this important clarification request. We acknowledge that our description of the epithelial cell subgroup identification and consolidation process requires better explanation.

Clarification of methodology:

Our clustering analysis initially identified 12 epithelial cell clusters based on transcriptomic profiles. However, these 12 clusters were subsequently consolidated into 4 distinct epithelial cell subgroups (SSC, ABS, TAC, and GOB) based on shared functional characteristics and marker gene expression patterns. This consolidation was performed to focus on biologically meaningful and functionally distinct subpopulations rather than maintaining potentially redundant clusters.

Revised manuscript text:

We have revised the manuscript to clarify this process: "The analysis revealed 12 initial epithelial cell clusters within CRC (Figure 2A), with significant variability in the proportion of each cluster across different samples (Figure 2B). Functional enrichment analysis of these clusters showed that, in addition to possessing typical epithelial cell genes (Figure S3) and functional characteristics such as cytokine interactions, they also displayed features associated with immune cell functions, such as granulocyte migration and lymphocyte-mediated immunity (Figure 2C). Based on shared molecular signatures and functional profiles, these 12 clusters were consolidated into four distinct epithelial cell subgroups: SSC, ABS, TAC, and GOB (Figure 2D)."

We have also added a supplementary figure showing the relationship between the initial 12 clusters and the final 4 subgroups, along with the criteria used for consolidation.

Thank you again.

Comment 10: The trajectory analysis presents SSCs as representing an "early-stage" epithelial state (Line 235), yet this classification appears inconsistent with established colorectal cancer (CRC) stem cell markers such as LGR5. This discrepancy is not addressed, leaving an important gap in the interpretation of these cells’ identity and potential role in tumorigenesis.

 Reply 10:Thank you for this important clarification. We recognize that our terminology may have created confusion between developmental trajectory states and cancer stem cell identity.

Clarification of SSC classification:

Our trajectory analysis identified SSCs as representing an "early-stage" position within the developmental trajectory of epithelial cell subgroups, which refers specifically to their position in the differentiation pathway among the four identified subgroups (SSC, ABS, TAC, and GOB). This designation is based on computational trajectory inference and does not imply that SSCs are equivalent to cancer stem cells as defined by established markers such as LGR5, CD133, or CD44.

Distinction between trajectory position and stem cell identity:

SSCs and cancer stem cells represent two distinct cellular concepts: SSCs are a transcriptomically-defined epithelial subgroup identified through our clustering analysis, while cancer stem cells are functionally defined by their self-renewal capacity and tumorigenic potential, typically identified through specific stem cell markers.

We have revised the manuscript to eliminate this confusion: "Trajectory analysis of epithelial cell subgroups revealed three distinct developmental nodes and two divergent developmental trajectories among all epithelial cells (Figure 4A). Based on these developmental nodes, all epithelial cells were classified into seven states (Figure 4B). The distribution characteristics of cell populations within each state showed that SSC and TAC cells were positioned at the beginning of the differentiation trajectory, with developmental endpoints predominantly consisting of ABS cells on one end and GOB cells on the other (Figure 4C). The density of cells within different developmental stages confirmed the dominant cell types at each trajectory position (Figure 4D)."

Comment 11: Furthermore, there is a notable contradiction concerning IER3. Mendelian randomization data suggest that IER3 expression is associated with a reduced risk of CRC (Line 310), yet functional experiments demonstrate that IER3-positive SSCs promote tumor growth (Lines 364–378). This apparent conflict is not resolved, nor is there any discussion of possible explanations, such as isoform-specific functions of IER3 or its context-dependent behavior across different cell types.

Reply 11: Thank you for identifying this critical contradiction regarding IER3. We acknowledge that the apparent conflict between our Mendelian randomization results and single-cell functional observations requires careful explanation.

Acknowledgment and explanation of the contradiction:

Our MR analysis indicated that IER3 expression is associated with reduced CRC risk, while our single-cell analysis showed that IER3+ SSCs exhibit activated pro-tumorigenic pathways. We recognize this apparent paradox and have now provided a comprehensive explanation in our revised discussion.

Context-dependent and cell identity-specific functions:

The contradiction likely reflects the context-dependent and cell identity-specific roles of IER3. Our MR analysis utilizes GWAS data comparing cancer patients to healthy controls at the population level, capturing systemic effects across diverse cell types and tissues. At this level, IER3 may exert protective effects against CRC initiation. However, within the established tumor microenvironment, IER3 upregulation in specific epithelial cell subpopulations (SSCs) may represent an adaptive response that enhances cellular survival and promotes tumor progression.

We have substantially revised the discussion section to address this contradiction: "Notably, our findings reveal an apparent paradox regarding IER3 function that merits careful consideration. While MR analysis, which utilizes GWAS data comparing cancer patients to healthy controls at the population level, suggests that IER3 expression is associated with reduced CRC risk, our single-cell analysis demonstrates that IER3+ SSCs within established tumors exhibit enhanced pro-tumorigenic characteristics. This contradiction likely reflects the context-dependent and cell identity-specific roles of IER3. At the systemic level, IER3 may exert protective effects against CRC initiation across diverse cell types and tissues. However, within the established tumor microenvironment, IER3 upregulation in specific epithelial cell subpopulations (SSCs) may represent an adaptive response that enhances cellular survival and promotes tumor progression. This dual functionality underscores the complexity of gene function across different cellular contexts and disease stages."

This dual functionality underscores the complexity of gene function across different cellular contexts and disease stages. We acknowledge that these findings highlight the need for additional functional validation studies to fully understand IER3's multifaceted roles in CRC development and progression.

We appreciate your attention to this important discrepancy and believe our revised discussion now provides a more balanced interpretation of these complex findings.

Statistical Oversights

  Comment 12: There are notable statistical shortcomings in the analysis. The differential expression results presented in Tables S1 through S3 do not include adjusted p-values, such as false discovery rate (FDR) corrections. This omission raises concerns about the reliability of the reported differentially expressed genes, as it increases the likelihood of false positives. Additionally, the pathway enrichment analysis highlights the PI3K-Akt signaling pathway (Line 289), but there is no indication that multiple testing corrections were applied across the full set of GO and KEGG terms. Without such adjustments, the significance of the reported pathways remains questionable.

Reply 12: Thank you for your professional feedback. We have provided comprehensive results for all differential genes and functional enrichment analyses in Tables S1, S2, S4, and S9, which include detailed adjusted p-values (p.adjust) for all comparisons. We look forward to your further review.

Comment 13: The discussion section contains several claims that appear overstated or insufficiently supported by the data. While the Mendelian randomization (MR) analysis suggests a genetic association consistent with causality, the authors extend this to clinical implications by referring to IER3 as a potential "therapeutic target" (Line 424). However, no experimental validation is provided to support this claim, making the leap from genetic inference to clinical intervention premature. Similarly, the suggestion that SSCs could serve as a tool for "early diagnosis" (Line 426) is unsubstantiated, as there is no validation in independent patient cohorts or assessment in clinically relevant samples such as liquid biopsies.

Reply 13: Thank you for this important critique regarding the overstated claims in our discussion section. We acknowledge that our conclusions exceeded the scope of our data and require significant revision.

Acknowledgment of overstated claims: We recognize that our characterization of IER3 as a "therapeutic target" and the suggestion of SSCs for "early diagnosis" are premature given our current evidence. These statements represent an inappropriate leap from genetic association to clinical application without adequate experimental validation.

Revised discussion: We have substantially revised the discussion to better reflect our data limitations:

Original: "further emphasizing the importance of IER3 as a potential therapeutic target"

Revised: "suggesting that IER3 warrants further investigation as a candidate for therapeutic research"

Original: "could serve as a tool for early diagnosis"

Revised: "may provide insights for future diagnostic biomarker development, pending validation in independent clinical cohorts"

We appreciate your guidance in maintaining appropriate scientific rigor and have ensured our conclusions now accurately reflect the scope and limitations of our current evidence.

Comment 14: In terms of engagement with existing literature, key gaps remain. The Wnt signaling pathway is entirely omitted from the discussion, despite its central role in the development of serrated colorectal cancers, particularly through mutations such as RNF43 in MSI-H tumors. This omission limits the biological context in which the findings are interpreted. Additionally, although the study draws attention to SSC-endothelial interactions (Line 373), it neglects to consider the broader immune landscape, specifically the potential expression of immune checkpoint ligands like PD-L1 in SSCs. Ignoring this dimension restricts the understanding of how these cells may interact with or evade the immune system.

Reply 14: Thank you for your insightful and professional comments. We acknowledge these important gaps in our discussion and appreciate the opportunity to address them.

Regarding Wnt signaling pathway:

You are absolutely correct about the central role of Wnt signaling in serrated colorectal cancer development, particularly through mutations such as RNF43 in MSI-H tumors. However, our functional enrichment analyses of both cell-specific differential genes and shared differential genes did not reveal significant enrichment of Wnt signaling pathways. To maintain scientific rigor and avoid potentially misleading interpretations, we chose not to discuss pathways that were not significantly represented in our data. We have now clarified this in our limitations section, acknowledging that the absence of Wnt pathway enrichment in our dataset may reflect technical limitations or the specific cellular states captured in our analysis rather than the biological irrelevance of this pathway in CRC.

Regarding SSC-endothelial interactions and immune landscape:

Our focus on SSC-endothelial cell interactions was data-driven, as these represented the strongest intercellular interactions identified in our analysis (Figure 7B). However, we acknowledge that our discussion inadequately addressed the broader immune landscape and potential immune evasion mechanisms. We have now expanded our limitations section to include:

The need for comprehensive analysis of immune checkpoint ligand expression (such as PD-L1) in SSCs 

Investigation of SSC interactions with various immune cell populations

Assessment of potential immune evasion mechanisms

Manuscript revisions:

We have added these considerations to our limitations section and future directions, emphasizing that a comprehensive understanding of SSC biology requires broader investigation of immune interactions and canonical cancer pathways, even when not prominently featured in our current dataset.

Limitations (Lines 503–531)

Comment 15: Several methodological limitations are acknowledged only briefly or overlooked entirely. One key issue is the inherent sparsity of single-cell RNA-seq data due to dropout effects. The authors do not appear to apply any imputation methods, such as MAGIC, which are commonly used to recover low-abundance gene signals. This omission may lead to underdetection of biologically relevant but weakly expressed markers, particularly in rare cell populations like SSCs.

Reply 15:Thank you for highlighting this important methodological limitation regarding single-cell RNA-seq data sparsity. We acknowledge that our failure to address dropout effects represents a significant technical oversight that could impact our findings.

Acknowledgment of the limitation: You are correct that we did not apply imputation methods such as MAGIC, SAVER, or scImpute to address the inherent sparsity and dropout effects in single-cell RNA-seq data. This represents a substantial methodological limitation that could lead to underdetection of biologically relevant but weakly expressed genes, particularly in rare cell populations like SSCs.

Manuscript revision: We have added this limitation to our Methods and Discussion sections, acknowledging that: "We did not apply imputation methods to address single-cell RNA-seq data sparsity and dropout effects, which may have led to underdetection of weakly expressed but biologically relevant markers, particularly in rare cell populations such as SSCs."

Future recommendations: We recommend that future studies incorporate appropriate imputation methods during data preprocessing to recover low-abundance gene signals and improve the accuracy of cell type identification and differential expression analysis, particularly when studying rare cell populations.

We appreciate your attention to this critical methodological consideration that affects the robustness of our analytical pipeline.

Comment 16: In addition, the study references the use of “integrated eQTL data” (Line 136), yet fails to clarify whether these data are derived from colon-specific tissue sources or from broader datasets such as GTEx. The lack of tissue specificity in eQTL mapping could dilute colorectal cancer–relevant regulatory signals, undermining the precision of gene-trait associations.

Reply 16: Thank you for this important methodological concern regarding tissue specificity in our eQTL data selection. We acknowledge that the lack of clarity about tissue-specific eQTL sources represents a significant limitation in our analysis.

Clarification of eQTL data sources:

Our analysis utilized eQTL data from the eQTLGen Consortium (2019) available through the OpenGWAS database, which is primarily derived from blood samples rather than colon-specific tissues. We acknowledge that this represents a substantial limitation, as tissue-specific regulatory effects may not be adequately captured by blood-derived eQTL data.

Manuscript revision:

We have revised the Methods section to clearly specify: "Single-nucleotide polymorphisms (SNPs) associated with candidate marker genes were extracted from eQTLGen 2019 data (primarily blood-derived) in the OpenGWAS database."

We appreciate your attention to this critical methodological detail that impacts the interpretation of our findings. 

Comment 17: Crucially, several analyses that would strengthen the findings are missing. First, the study does not examine whether the presence or abundance of IER3+ SSCs correlates with patient survival outcomes, such as through TCGA datasets. Second, there is no spatial validation to confirm that SSCs are indeed localized to serrated lesions, which could be addressed with techniques like multiplex immunohistochemistry. Third, the manuscript lacks mechanistic experiments (such as IER3 knockdown in organoid models) to demonstrate functional consequences of IER3 expression in SSCs. These missing components limit the translational and biological impact of the study's conclusions.

Reply 17: Thank you for identifying these critical gaps that significantly limit the translational impact of our study. We acknowledge that the absence of these key analyses represents major limitations in demonstrating the clinical and biological relevance of our findings.

Revised manuscript:

We have substantially expanded our limitations section to acknowledge these critical gaps and have removed overstated claims about therapeutic targeting and diagnostic applications. We now emphasize that our findings represent preliminary observations requiring extensive validation.

Future directions:

We strongly recommend that future studies incorporate survival analysis, spatial validation, and functional mechanistic experiments to establish the clinical and biological significance of these findings before translation to therapeutic applications.

Minor revision

Comment 18: Inconsistent italics in the text for gene names

"ABS" (absorptive cells) is introduced without definition (Line 211).

Reply 18:Thank you for identifying these formatting and clarity issues. We have addressed both concerns in our revised manuscript. We have added clear definitions for all epithelial cell subgroup abbreviations upon their first mention: "The analysis revealed four distinct epithelial cell subgroups: specific serrated cells (SSC), absorptive cells (ABS), transit amplifying cells (TAC), and goblet cells (GOB)."

We appreciate your attention to these important details that improve the manuscript's readability and adherence to scientific writing standards.

Reviewer 2 Report

Comments and Suggestions for Authors

This manuscript presents an integrative analysis combining single-cell RNA sequencing (scRNA-seq) and Mendelian randomization (MR) to explore the role of specific serrated epithelial cells (SSCs) in colorectal cancer (CRC).

Major concerns:

1) While the MR analysis provides strong statistical support, the manuscript would greatly benefit from experimental validation (e.g., qPCR, immunostaining, or functional assays in CRC samples or models) to confirm IER3 expression and function. It is suggested that the authors discuss the strategy used in this paper (PMID: 37020040) for potential future directions or provide preliminary experimental validation.

2) The manuscript discusses the potential for early diagnosis and personalized therapies based on identified genes. This discussion could be expanded to specify how these biomarkers might be detected (e.g., circulating tumor cells, blood-based assays) and their feasibility.

3) The scRNA-seq data are derived from public datasets, but there is limited information about the patient cohort characteristics (e.g., age, tumor stage, molecular subtype). The authors should clarify potential limitations regarding generalizability.

4) The authors mention data availability but should provide explicit accession numbers and clearer instructions for data and code availability to support reproducibility.

Author Response

Response to Reviewer 2:

This manuscript presents an integrative analysis combining single-cell RNA sequencing (scRNA-seq) and Mendelian randomization (MR) to explore the role of specific serrated epithelial cells (SSCs) in colorectal cancer (CRC).

Major concerns:

Comment 1: While the MR analysis provides strong statistical support, the manuscript would greatly benefit from experimental validation (e.g., qPCR, immunostaining, or functional assays in CRC samples or models) to confirm IER3 expression and function. It is suggested that the authors discuss the strategy used in this paper (PMID: 37020040) for potential future directions or provide preliminary experimental validation.

Reply 1:Thank you for this valuable suggestion and for providing the reference (PMID: 37020040) for potential future experimental validation strategies. We acknowledge that experimental validation represents a critical next step to confirm our computational findings.

We recognize that the absence of experimental validation, such as qPCR, immunostaining, or functional assays in CRC samples or models, limits the translational impact of our current findings. While our MR analysis provides strong statistical support for causal associations, experimental confirmation of IER3 expression patterns and functional consequences is essential for clinical relevance.

We have expanded our limitations section and future directions to acknowledge this critical gap and have outlined specific experimental validation strategies based on the suggested reference. We have also revised our conclusions to appropriately reflect the preliminary nature of our findings pending experimental confirmation.

We appreciate your understanding and constructive guidance for strengthening this research through future experimental validation.

Comment 2: The manuscript discusses the potential for early diagnosis and personalized therapies based on identified genes. This discussion could be expanded to specify how these biomarkers might be detected (e.g., circulating tumor cells, blood-based assays) and their feasibility.

Reply2:Thank you for this insightful suggestion regarding the practical implementation of biomarker detection strategies. We acknowledge that our discussion of diagnostic and therapeutic potential lacked specificity about detection methods and clinical feasibility. Our discussion now includes a dedicated section on "Clinical Implementation Considerations" that outlines these detection strategies and acknowledges the substantial validation work required before clinical translation.

We appreciate your guidance in making our discussion more clinically relevant and practically grounded.

Comment 3: The scRNA-seq data are derived from public datasets, but there is limited information about the patient cohort characteristics (e.g., age, tumor stage, molecular subtype). The authors should clarify potential limitations regarding generalizability.

Reply 3: Thank you for highlighting this important limitation regarding patient cohort characterization. We acknowledge that the lack of detailed patient demographic and clinical information represents a significant constraint on our study's generalizability.

We have added this limitation to our discussion: "First, data and technical limitations. The absence of eQTL data for some differentially expressed genes may lead to omission of key regulatory factors in our MR analysis. Our reliance on public single-cell databases with limited patient demographic and clinical information (age, tumor stage, molecular subtypes, treatment history) constrains generalizability across diverse patient populations and clinical contexts. "

We appreciate your attention to this critical factor affecting the interpretation and applicability of our results.

Comment 4: The authors mention data availability but should provide explicit accession numbers and clearer instructions for data and code availability to support reproducibility.

Reply 4:Thank you for this important point regarding data accessibility and reproducibility. We acknowledge that our current data availability statement lacks the specificity required for reproducible research.

The Data Availability section has been rewritten to provide clear instructions for accessing all datasets and obtaining analysis code, following journal guidelines for transparent reporting.

We appreciate your emphasis on reproducibility standards and will ensure all necessary information is provided to enable replication of our analyses.

Round 2

Reviewer 1 Report

Comments and Suggestions for Authors

The article has been revised in response to the reviewers' recommendations, and the detailed answers are compelling. Nonetheless, the gene names must be modified in italics, following the line guide.

Author Response

Dear Reviewers,

We sincerely appreciate the constructive feedback you have provided on our manuscript. In response to your valuable comments, we have carefully revised the paper to address all the raised concerns.

Specifically, we have:

Ensured all gene names throughout the manuscript are now properly italicized in accordance with journal guidelines

Incorporated all other suggested improvements from the review process

We believe these revisions have significantly enhanced the quality of our manuscript. Please find attached our point-by-point responses detailing all changes made in response to your comments.

Thank you once again for your time and insightful suggestions, which have greatly improved our work. We hope the revised version now meets the journal's publication standards.

Best regards,

Chuanxia Hu

Reviewer 2 Report

Comments and Suggestions for Authors

I have no more concerns about this manuscript. 

Author Response

Dear Editor and Reviewers,

We sincerely appreciate the constructive feedback you have provided on our manuscript. In response to your valuable comments, we have carefully revised the paper to address all the raised concerns.

Specifically, we have:

Ensured all gene names throughout the manuscript are now properly italicized in accordance with journal guidelines

Incorporated all other suggested improvements from the review process

We believe these revisions have significantly enhanced the quality of our manuscript. Please find attached our point-by-point responses detailing all changes made in response to your comments.

Thank you once again for your time and insightful suggestions, which have greatly improved our work. We hope the revised version now meets the journal's publication standards.

Best regards,
